# Ectopic Fat Accumulation in Pancreas and Heart

**DOI:** 10.3390/jcm10061326

**Published:** 2021-03-23

**Authors:** Junji Kozawa, Iichiro Shimomura

**Affiliations:** 1Department of Metabolic Medicine, Graduate School of Medicine, Osaka University, Suita 565-0871, Japan; ichi@endmet.med.osaka-u.ac.jp; 2Department of Diabetes Care Medicine, Graduate School of Medicine, Osaka University, Suita 565-0871, Japan

**Keywords:** ectopic fat accumulation, diabetes, pancreas, heart

## Abstract

Ectopic fat is found in liver, muscle, and kidney and is known to accumulate as visceral fat. In recent years, ectopic fat has also been observed in the pancreas, and it has been said that pancreatic fat accumulation is related to the pathophysiology of diabetes and the onset of diabetes, but the relationship has not yet been determined. In the heart, epicardium fat is another ectopic fat, which is associated with the development of coronary artery disease. Ectopic fat is also observed in the myocardium, and diabetic patients have more fat accumulation in this tissue than nondiabetic patients. Myocardium fat is reported to be related to diastolic cardiac dysfunction, which is one of the characteristics of the complications observed in diabetic patients. We recently reported that ectopic fat accumulation was observed in coronary arteries of a type 2 diabetic patient with intractable coronary artery disease, and coronary artery is attracting attention as a new tissue of ectopic fat accumulation. Here, we summarize the latest findings focusing on the relationship between ectopic fat accumulation in these organs and diabetic pathophysiology and complications, then describe the possibility of future treatments targeting these ectopic fat accumulations.

## 1. Introduction

Ectopic fat is a fat accumulation in or around specific organs or compartments of the body. The liver is a typical organ that causes ectopic fat and is known to be deeply involved in the pathophysiology of diabetes [1,2,3]. In recent years, it has been elucidated that fat accumulations are also observed in organs such as skeletal muscle, kidney, heart, and pancreas [4]. These fat accumulations have been discussed in relation to diabetes [4]. Skeletal muscle is an organ that is responsible for postprandial insulin-stimulated glucose disposal [5]. Intramuscular lipid is associated with impaired glucose uptake in skeletal muscle in insulin-resistant subjects [6]. However, the skeletal muscle of trained athletes with elevated lipid content is significantly insulin-sensitive, which is a phenomenon known as the athlete’s paradox [7], so the effect of intramuscular fat accumulation on glucose metabolism may still be controversial. Fat accumulation in the kidney is mainly observed in the renal sinus [8]. This fat accumulation is associated with an increased risk of hypertension [9] and chronic kidney disease [8,10,11]. Ectopic fat depositions in the pancreas and heart are now highlighted, but the effects of these fat accumulations on organ-specific function and their pathophysiological significance are unknown. The present paper reviewed clinical reports mainly examining the relationships between ectopic fat accumulations in the pancreas and heart and diabetic pathophysiology and complications.

## 2. Ectopic Fat in Pancreas

Ectopic fat is observed in the pancreas. The pancreas is roughly divided into pancreatic islets that secrete endocrine hormones such as insulin and glucagon, and exocrine regions that secrete digestive enzymes, and is composed of lobes separated by connective tissues. Pancreatic fat includes interlobular or intralobular infiltration of adipocytes [12] as well as accumulation of intracellular lipid droplets of pancreatic endocrine or exocrine cells [13,14,15]. So far, it has been said that pancreatic fat increases physiologically with age, obesity, diabetes mellitus, excess alcohol intake, and viral infections [14,16,17]. Fat accumulation in the pancreas is called pancreatic steatosis, fatty pancreas, etc. Recently, nonalcoholic fatty pancreas disease (NAFPD) has been proposed as a disease concept related to obesity in people who have never drunk [18]. 

There are various reports on the pathophysiological significance of this fat deposition, and it has not been determined whether this fat accumulation has a negative effect on glucose metabolism or not. We investigated pancreatic fat-cell infiltration in nondiabetic patients undergoing pancreatectomy. We found that fat-cell infiltration was associated with postoperative deterioration of glucose tolerance [19], and that this infiltration in addition to hyperglycemia was also associated with islet inflammation, which was evaluated by macrophage infiltration around or within islets [20]. Furthermore, it has been clarified that pancreatic fat evaluated by computed tomography value is involved in the decrease in endogenous insulin-secreting capacity in type 2 diabetic patients [21]. According to all these reports, fat-cell infiltration might be involved in the deterioration of the insulin-secreting capacity through islet inflammation. 

The mRNA of perilipin 2, which is a lipid droplet constituent protein, is increased in the pancreas of type 2 diabetic patients, and its protein expression is also increased in beta cells, which has been reported to be associated with impaired autophagy in beta cells [13]. Furthermore, it has been reported that there is a disorder of lipolysis in pancreatic islets of type 2 diabetic patients, and that inhibition of adipose triglyceride lipase (ATGL), which is a lipolytic enzyme, increases the size of lipid droplets and induces the deterioration of glucose-responsive insulin secretion [15], suggesting that intracellular lipid accumulation might also be associated with beta-cell dysfunction, resulting in glucose intolerance (Figure 1).

According to the reports described above, pancreatic fat accumulation might be strongly associated with the pathophysiology and the onset of diabetes. Therefore, it is considered that reducing pancreatic fat accumulation might have the possibility of the improvement of diabetic pathophysiology, resulting in the prevention of the onset or deterioration of diabetes. Some anti-diabetic drugs, which include glucagon-like peptide-1 receptor agonists (GLP-1RA) [22,23], sodium-glucose cotransporter-2 (SGLT-2) inhibitors [24,25], and PPARα agonists [26], have been reported to have an effect of reducing hepatic fat accumulation in human or rodent models of nonalcoholic steatohepatitis (NASH). These drugs have been shown not only to improve the pathophysiology of fatty liver disease but also to prevent liver carcinogenesis in NASH model mice [27]. In the pancreas, there are some reports suggesting a relationship between pancreatic fat accumulation and pancreatic cancer [28,29]. Considering these reports, GLP-1RA, SGLT-2 inhibitors, and PPARα agonists might have the possibility that they also reduce pancreatic fat accumulation, resulting in the improvement of diabetic pathophysiology and pancreatic carcinogenesis. We enrolled 22 type 2 diabetic outpatients, who had undergone unenhanced abdominal CT examinations before and more than 3 months after SGLT-2 inhibitor administration, and investigated the changes of the degrees of liver and pancreatic fat accumulations [30]. As a result, no improvement in these fat accumulations was observed in total. However, when restricted to the patients with intense cumulative fat accumulations in these organs, each fat accumulation was reduced after SGLT-2 inhibitor treatment. In addition, interestingly, the patients with intense pancreatic fat deposition were not always the same as those with intense liver fat accumulation. The degree of liver fat accumulation showed strong correlations with obesity-related parameters, while the degree of pancreatic fat accumulation was weakly or not associated with these parameters. These results suggest that pancreatic fat accumulation has a different etiology from liver fat accumulation. It is needed to elucidate the mechanism of the improvement of pancreatic fat accumulation by SGLT-2 inhibitor administration and to verify it in prospective clinical studies. 

## 3. Ectopic Fat in Heart 

Epicardial fat is known as ectopic fat in the heart. Epicardial fat is the fat that exists between the parietal and visceral pericardium and has a protective role for the heart, such as the source of free fatty acids, which is the energy for cardiomyocytes, the physical protection of coronary arteries and the heat retention effect [31]. On the other hand, it has been reported that excess epicardial fat is involved in lipotoxicity due to supplying excess free fatty acids, cardiac hypertrophy, and cardiac diastolic dysfunction [31]. Epicardial fat is increased in patients with type 2 diabetes [30,31], and its accumulation has been reported to be associated with the presence of coronary artery disease [32,33]. 

A genetic mutation of adipose triglyceride lipase (ATGL) causes accumulation of triglyceride not only in the myocardium but also in the coronary arteries, resulting in intractable heart failure, arrhythmia, and coronary artery lesions. This phenotype was reported as triglyceride deposit cardiomyovasculopathy (TGCV) in a cardiac transplant recipient [34]. Apart from this rare disease, the ectopic fat accumulation in the heart of diabetic patients has been discussed. It has become possible to estimate fat accumulation in the myocardium using proton magnetic resonance (MR) spectroscopy [35]. This modality revealed that type 2 diabetic patients had increased intramyocardial fat mass compared to healthy subjects, which is associated with myocardial diastolic dysfunction [36]. Diastolic function in type 2 diabetic patients is deteriorated compared with type 1 diabetic patients [37], while patients with both types of diabetes have abnormal resting heart high-energy phosphates (HEPs) [38,39]. Lipid accumulation, in addition to this abnormal HEPs metabolism in type 2 diabetic patients, might worsen cardiac function. Furthermore, this increased fat accumulation in the myocardium is observed regardless of the presence or absence of obesity [40]. Coronary artery disease in diabetic patients is characterized by diffuse narrowing and multi-vessel lesions [41,42] and frequent restenosis after treatment [43], but its true nature has not been clarified. We experienced a type 2 diabetic male patient, who presented with obesity and metabolic syndrome, underwent frequent catheter interventions and cardiac bypass surgery due to refractory heart failure, arrhythmia, and coronary artery disease, and subsequently died [44]. When analyzing the coronary artery wall by computed tomography (CT) value, there were regions with low CT values mainly located in the media and intima of the coronary arteries, suggesting the presence of triglyceride accumulation [45]. According to this case, it was suggested that intractable coronary artery disease observed in diabetic patients might be due to triglyceride accumulation centered on the media region of the coronary wall, unlike conventional atherosclerosis characterized by cholesterol accumulation [46,47]. As another form of ectopic fat deposition, nonalcoholic steatohepatitis (NASH) aggravates nitric oxide synthase (NOS) inhibition-induced arteriosclerosis in NASH model rats [48]. It is possible that ectopic fat accumulation in coronary arteries might also cause the dysregulation of NOS, which is inhibited by asymmetric dimethylarginine [49], resulting in coronary artery disease. In addition to the previously known epicardium and myocardium fat accumulations, ectopic fat accumulation in the coronary arteries might be linked to the pathophysiology of intractable heart disease in diabetic patients (Figure 2), and is considered as a new therapeutic target. It has been reported that the thickness of epicardial fat decreased significantly after administration of canagliflozin, an SGTT-2 inhibitor, to Japanese type 2 diabetic patients [50]. This report suggests that SGLT-2 inhibitors might have an effect of reducing ectopic fat accumulation not only in the liver and pancreas but also in the epicardial fat. Future studies will be needed to determine whether not only SGLT-2 inhibitors but also PPARα agonists, which are expected to reduce ectopic fat accumulation, can also reduce ectopic fat accumulations in the myocardium and coronary arteries.

## 4. Conclusions

Ectopic fat accumulation is found in the pancreas and heart, but factors other than obesity associated with these accumulations have not been clarified. Ectopic fat in the pancreas is associated with diabetic pathophysiology and glycemic control, and fat accumulation in the heart, including coronary arteries, is associated with intractable heart disease. It is necessary to clarify the factors that cause these fat accumulations and seek treatment modalities that contribute to their reduction.

## Figures and Tables

**Figure 1 jcm-10-01326-f001:**
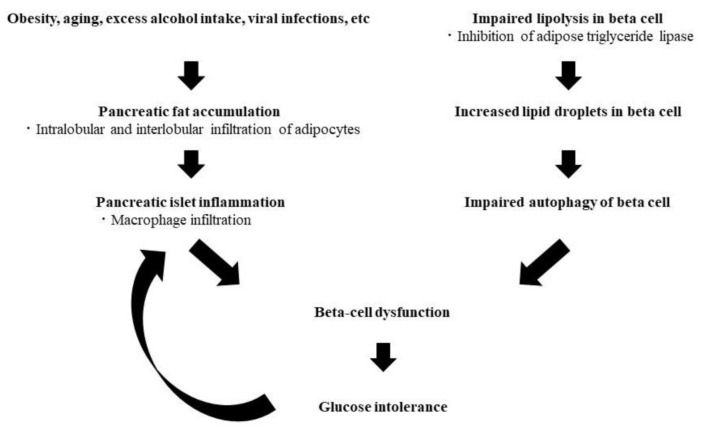
Hypothesis of the pathophysiology of glucose intolerance derived from ectopic fat accumulation in the pancreas.

**Figure 2 jcm-10-01326-f002:**
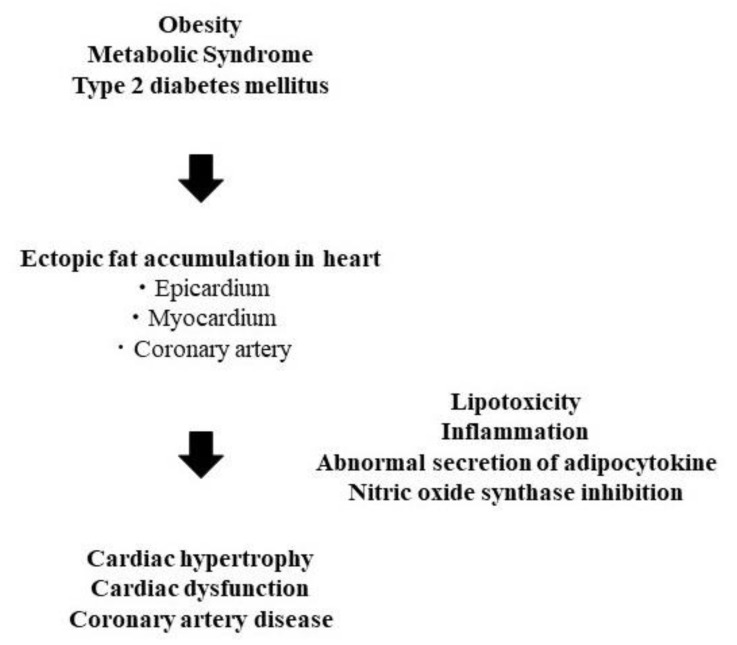
Hypothesis of the pathophysiology of cardiac disease derived from ectopic fat accumulation in heart.

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
