# Peer review of "Ectopic Fat Accumulation in Pancreas and Heart"

_jcm, 2021, doi:10.3390/jcm10061326_

Round 1
Reviewer 1 Report
this paper is very informative.
However, it is not well organized.
The author needs to improve the quality of the Figures. Figures looks like preliminary data.
Author need to reorganization of all papers.
It is recommended that the introduction is summarized in general fat-related information. As well as the pancreas-related information is collected in the pancreas part, and the heart-related information is collected in the heart part.
This is because the same information comes out too much repeatedly
Author Response
We are grateful to the reviewers for helpful comments and suggestions.
We responded to them point by point as follows.
Comments from Reviewer 1:
Point 1ï¼›The author needs to improve the quality of the Figures. Figures looks like preliminary data.
Response 1ï¼›
Thank you for the reviewer’s valuable and important comments. As the reviewer indicated, I corrected the Figure 1 and 2. I added some explanations.
Point 2ï¼›Author need to reorganization of all papers. It is recommended that the introduction is summarized in general fat-related information. As well as the pancreas-related information is collected in the pancreas part, and the heart-related information is collected in the heart part. This is because the same information comes out too much repeatedly.
Response 2ï¼›
Thank you for the reviewer’s valuable and important comments. As the reviewer suggested, I summarized general fat-related information in “Introduction”, then wrote the pancreas-related information in the pancreas part and the heart-related part in the heart part. I corrected the introduction as follows and added the references 5-11. I reorganized all papers and renumbered them.
-Line 31-43:“These fat accumulations have been discussed in relation to diabetes [4]. Skeletal muscle is an organ which is responsible for postprandial insulin-stimulated glucose disposal [5]. Intramuscular lipid is associated with impaired glucose uptake in skeletal muscle in insulin-resistant subjects {6}. However, skeletal muscle of trained athletes with elevated lipid content is significantly insulin-sensitive, which is a phenomenon known as the athlete’s paradox [7], so the effect of intramuscular fat accumulation on glucose metabolism may be still controversial. Fat accumulation in kidney is mainly observed in the renal sinus [8]. This fat accumulation is associated with increased risk of hypertension [9] and chronic kidney disease [8, 10,11]. Ectopic fat depositions in pancreas and heart are now highlighted, but the effects of these fat accumulations on organ-specific function and their pathophysiological significance are unknown. The present paper reviewed clinical reports mainly examining the relationships between ectopic fat accumulations in pancreas and heart and diabetic pathophysiology and complications.” has been added.
-Line 204-217:
“5. Baron AD, Brechtel G, Wallace P, Edelman SV. Rates and tissue sites of non-insulin- and insulin-mediated glucose uptake in humans. Am J Physiol. 1988, Dec;255(6 Pt 1):E769-774.”,
“6. Jacob S, Machann J, Rett K, Brechtel K, Volk A, Renn W, Maerker E, Matthaei S, Schick F, Claussen CD, Häring HU. Association of increased intramyocellular lipid content with insulin resistance in lean nondiabetic offspring of type 2 diabetic subjects. Diabetes. 1999 May;48(5):1113-1119.”,
“7. Goodpaster BH, He J, Watkins S, Kelley DE. Skeletal muscle lipid content and insulin resistance: evidence for a paradox in endurance-trained athletes. J Clin Endocrinol Metab. 2001 Dec;86(12):5755-5761.”,
“8. Foster MC, Hwang SJ, Porter SA, Massaro JM, Hoffmann U, Fox CS. Fatty kidney, hypertension, and chronic kidney disease: the Framingham Heart Study. Hypertension. 2011 Nov;58(5):784-790.”,
“9. Chughtai HL, Morgan TM, Rocco M, Stacey B, Brinkley TE, Ding J, Nicklas B, Hamilton C, Hundley WG. Renal sinus fat and poor blood pressure control in middle-aged and elderly individuals at risk for cardiovascular events. Hypertension. 2010;56:901–906.”,
“10. Irazabal MV, Eirin A. Role of Renal Sinus Adipose Tissue in Obesity-induced Renal Injury. EBioMedicine. 2016 Nov;13:21-22.”,
“11. Shen FC, Cheng BC, Chen JF. Peri-renal fat thickness is positively associated with the urine albumin excretion rate in patients with type 2 diabetes. Obes Res Clin Pract. 2020 Jul-Aug;14(4):345-349.” have been added.
-Line 43-46ï¼›“Fat accumulation in pancreas is called pancreatic steatosis, fatty pancreas, etc. Recently, non-alcoholic fatty pancreas disease (NAFPD) has been proposed as a disease concept related to obesity in people who have never drunk [5]” has been moved to Line 69-72.
-Line 46-61:“Although it has been suggested that ectopic fat in these organs is associated with visceral fat accumulation [6], it is unclear whether there are additional specific factors that cause fat accumulation in pancreas. More recently, the presence of lipid droplets in pancreatic beta cell has been reported [7-9], and it is suggested to be related to diabetic conditions such as deteriorated insulin-secreting capacity and the onset of diabetes. In heart, epicardial fat is the fat that is originally present between myocardium and visceral layer of the pericardium and exerts metabolic, thermogenic and mechanical cardioprotective properties [10]. However, when it accumulates excessively, it is known to be involved in the development of coronary artery disease [11,12]. In addition, it has been found that fat accumulation in myocardium is observed, and that diabetic patients have more fat accumulation than non-diabetic patients and are involved in diabetic disorders such as cardiac diastolic dysfunction [13]. Furthermore, coronary artery disease in diabetes patients is characterized by intractable coronary artery disease that presents with diffuse narrowing regions and a high rate of restenosis after treatment. It is attracting attention that such intractable coronary artery disease in diabetes patients might be associated with ectopic fat accumulation in the coronary [14].” have been deleted.
In addition to these changes, I have corrected the following sentence in Line 29-30.
-Line 29-30ï¼›“fat accumulation is also observed” has been replaced by “fat accumulations are also observed”.

Reviewer 2 Report
Dear Authors,
Overall the manuscript is clear and well written. The study is well designed.
Few flaws need to be addressed before considering the manuscript for publication.
Minor flaws:
- Line 16: Change “diabetes patients” with “diabetic patients”.
- To improve the discussion an interesting paper to discuss could be (Perseghin et al, J of Am College of Cardiol., 2005) where authors investigate whether left ventricular dysfunction was associated with abnormal heart high energy phosphates (HEPs) in type 1 diabetic patients. It will be interesting also to discuss this correlation in a type 2 diabetes scenario (Freestone et al, Circulation, 2007).
- Line 119 and 122 change “ diabetes patients” with “diabetic patients”.
- In the perspective of MR spectroscopy used to estimate fat accumulation in heart of diabetic patients, could be also interesting to discuss the manuscript of “Astorri et al, Cardiology, 1997” where authors used radionuclide angiography to assess left ventricular function in non-insulin-dependent an in insulin-dependent patients.
- Since a decrease of NO biodisponibility is associated with an increase of homocysteine and LDL levels and oxidative stres in diabetic patients with cardiovascular complications, an interesting paper to discuss could be “Paroni et al, Amino Acids, 2005”
- Line 134 change “ diabetes patients” with “diabetic patients”
Author Response
We are grateful to the reviewers for helpful comments and suggestions.
We responded to them point by point as follows.
Comments from Reviewer 2:
Point 1ï¼›To improve the discussion an interesting paper to discuss could be (Perseghin et al, J of Am College of Cardiol., 2005) where authors investigate whether left ventricular dysfunction was associated with abnormal heart high energy phosphates (HEPs) in type 1 diabetic patients. It will be interesting also to discuss this correlation in a type 2 diabetes scenario (Freestone et al, Circulation, 2007).
Response 1ï¼›Thank you for the reviewer’s valuable and important comments. As the reviewer comments, abnormal heart high-energy phosphates contribute to cardiac dysfunction in both type 1 and type 2 diabetic patients. I think that ectopic fat accumulation might additionally worsen cardiac function. I quoted 2 previous papers as the reviewer suggested, which reported abnormal resting heart high-energy phosphates (HEPs) in both type 1 and type 2 diabetic patients (Reference 38,39). Unfortunately, I could not find the paper “Freestone, et al. Circulation, 2007”, so I have cited reference 39 instead.
-Line 142-144ï¼›“while patients with both types of diabetes have abnormal resting heart high-energy phosphates (HEPs) [38,39]. Lipid accumulation in addition to this abnormal HEPs metabolism in type 2 diabetic patients might worsen cardiac function.” have been added.
-Line 283-287:
“38. Perseghin G, Fiorina P, De Cobelli F, Scifo P, Esposito A, Canu T, Danna M, Gremizzi C, Secchi A, Luzi L, Del Maschio A. Cross-sectional assessment of the effect of kidney and kidney-pancreas transplantation on resting left ventricular energy metabolism in type 1 diabetic-uremic patients: a phosphorous-31 magnetic resonance spectroscopy study. J Am Coll Cardiol. 2005 Sep 20;46(6):1085-1092.”,
“39. Scheuermann-Freestone M, Madsen PL, Manners D, Blamire AM, Buckingham RE, Styles P, Radda GK, Neubauer S, Clarke K. Abnormal cardiac and skeletal muscle energy metabolism in patients with type 2 diabetes. Circulation. 2003 Jun 24;107(24):3040-3046” have been added.
Point 2ï¼›In the perspective of MR spectroscopy used to estimate fat accumulation in heart of diabetic patients, could be also interesting to discuss the manuscript of “Astorri et al, Cardiology, 1997” where authors used radionuclide angiography to assess left ventricular function in non-insulin-dependent an in insulin-dependent patients.
Response 2ï¼›Thank you for the reviewer’s valuable and important comments. I quoted a previous paper as the reviewer 1 suggested (reference 37), which compared cardiac diastolic function between type 1 and type 2 diabetic patients.
-Line 140-142ï¼›“Diastolic function in type 2 diabetic patients is deteriorated compared with type 1 diabetic patients [37]” has been added.
-Line 281-282:
“37. Astorri E, Fiorina P, Gavaruzzi G, Astorri A, Magnati G. Left ventricular function in insulin-dependent and in non-insulin-dependent diabetic patients: radionuclide assessment. Cardiology. 1997 Mar-Apr;88(2):152-155.” has been added.
Point 3ï¼›Since a decrease of NO biodisponibility is associated with an increase of homocysteine and LDL levels and oxidative stres in diabetic patients with cardiovascular complications, an interesting paper to discuss could be “Paroni et al, Amino Acids, 2005”
Response 3ï¼›Thank you for the reviewer’s valuable and important comments. It is unknown whether ectopic fat accumulation in heart leads to the inhibition of nitric oxide synthase (NOS). However, in non-alcoholic steatohepatitis (NASH) model rats, NASH aggravates NOS inhibition-induced arteriosclerosis (reference 48). In reference to this report, it is possible that ectopic fat accumulation in coronary arteries might also cause the dysregulation of NOS, which is inhibited by asymmetric dimethylarginine (reference 49), resulting in coronary artery disease. We added the following sentences.
-Line 157-162ï¼›“As another form of ectopic fat deposition, non-alcoholic steatohepatitis (NASH) aggravates nitric oxide synthase (NOS) inhibition-induced arteriosclerosis in NASH model rats [48]. It is possible that ectopic fat accumulation in coronary arteries might also cause the dysregulation of NOS, which is inhibited by asymmetric dimethylarginine [49], resulting in coronary artery disease.”have been added.
-Line 305-308ï¼›
“48. Watanabe S, Kumazaki S, Yamamoto S, Sato I, Kitamori K, Mori M, Yamori Y, Hirohata S. Non-alcoholic steatohepatitis aggravates nitric oxide synthase inhibition-induced arteriosclerosis in SHRSP5/Dmcr rat model. Int J Exp Pathol. 2018 Dec;99(6):282-294.”
“49. Paroni R, Fermo I, Fiorina P, Cighetti G. Determination of asymmetric and symmetric dimethylarginines in plasma of hyperhomocysteinemic subjects. Amino Acids. 2005 Jun;28(4):389-394.” have been added.
Point 4ï¼›Line 16: Change “diabetes patients” with “diabetic patients”. Line 119 and 122 change “ diabetes patients” with “diabetic patients”. Line 134 change “ diabetes patients” with “diabetic patients”.
Response 4ï¼›Thank you for the reviewer’s helpful comments. I changed “diabetes patients” to “diabetic patients” in Line 16,18,19,81,85,88,110,136,139,146,148,155,164 and 167 in the revised file.

Round 2
Reviewer 1 Report
Improved manuscript.